# Novel 3D Bioglass Scaffolds for Bone Tissue Regeneration

**DOI:** 10.3390/polym14030445

**Published:** 2022-01-22

**Authors:** Evangelos Daskalakis, Boyang Huang, Cian Vyas, Anil Ahmet Acar, Ali Fallah, Glen Cooper, Andrew Weightman, Bahattin Koc, Gordon Blunn, Paulo Bartolo

**Affiliations:** 1Department of Mechanical, Aerospace and Civil Engineering, University of Manchester, Manchester M13 9PL, UK; evangelos.daskalakis@manchester.ac.uk (E.D.); boyang.huang@manchester.ac.uk (B.H.); cian.vyas@manchester.ac.uk (C.V.); glen.cooper@manchester.ac.uk (G.C.); andrew.weightman@manchester.ac.uk (A.W.); 2Integrated Manufacturing Technologies Research and Application Center, Sabanci University, Tuzla, Istanbul 34956, Turkey; aacar@sabanciuniv.edu (A.A.A.); ali.fallah@sabanciuniv.edu (A.F.); bahattinkoc@sabanciuniv.edu (B.K.); 3SUNUM Nanotechnology Research Center, Sabanci University, Tuzla, Istanbul 34956, Turkey; 4Faculty of Engineering and Natural Sciences, Sabanci University, Tuzla, Istanbul 34956, Turkey; 5School of Pharmacy and Biomedical Sciences, University of Portsmouth, Portsmouth PO1 2DT, UK; gordon.blunn@port.ac.uk; 6Singapore Centre for 3D Printing, School of Mechanical and Aerospace Engineering, Nanyang Technological University, Singapore 639798, Singapore

**Keywords:** 3D printing, bioglass, bone scaffolds, PCL, tissue engineering

## Abstract

The design of scaffolds with optimal biomechanical properties for load-bearing applications is an important topic of research. Most studies have addressed this problem by focusing on the material composition and not on the coupled effect between the material composition and the scaffold architecture. Polymer–bioglass scaffolds have been investigated due to the excellent bioactivity properties of bioglass, which release ions that activate osteogenesis. However, material preparation methods usually require the use of organic solvents that induce surface modifications on the bioglass particles, compromising the adhesion with the polymeric material thus compromising mechanical properties. In this paper, we used a simple melt blending approach to produce polycaprolactone/bioglass pellets to construct scaffolds with pore size gradient. The results show that the addition of bioglass particles improved the mechanical properties of the scaffolds and, due to the selected architecture, all scaffolds presented mechanical properties in the cortical bone region. Moreover, the addition of bioglass indicated a positive long-term effect on the biological performance of the scaffolds. The pore size gradient also induced a cell spreading gradient.

## 1. Introduction

The use of additive manufacturing techniques, also known as 3D printing, for the fabrication of customised bone tissue engineering architectures has developed very rapidly [1,2,3]. These fabrication techniques allow high reproducibility and repeatability, and they can process a wide range of natural and synthetic polymeric and composite materials, allowing for the fabrication of interconnected porous structures [4,5,6,7]. Different techniques have been used for the fabrication of bone scaffolds [8,9,10,11,12], such as: vat photopolymerisation, which uses light to selectively solidify a liquid photo-sensitive polymeric material; powder-bed fusion, which uses a laser beam to selectively fuse powder material; and extrusion-based processes, which heat materials in a pellet or filament form to solidify the molten material in a printing platform. Among these techniques, extrusion-based additive manufacturing is the most commonly used due to its low cost, high simplicity and ability to print multiple materials [13,14,15]. This technique comprises pneumatic systems using air pressure as the dispensing force, solenoid-based systems using electromagnetic forces and mechanical systems based on the use of a piston or screw [16]. The use of screw-assisted extrusion-based systems is the ideal technique to process high viscosity composite materials [17].

Polycaprolactone has been extensively investigated for use in bone scaffolds [18,19]. It is a semi-crystalline, biocompatible, biodegradable and hydrophobic synthetic polymer that presents good physicochemical characteristics and slow degradation kinetics [20,21]. Due to its low melting temperature, PCL is a suitable material to be printed using extrusion-based additive manufacturing [22]. To improve bioactivity, different concentrations of hydroxyapatite (HA) and tricalcium phosphate (TCP) have been added to PCL scaffolds [23,24,25,26,27,28]. This improves both the mechanical and hydrophilicity properties and accelerates mineralisation [29,30,31,32,33,34,35]. However, these scaffolds still present limitations in terms of mechanical properties and slow degradation. To decrease hydrophobicity and thus increase cell attachment, strategies such as plasma and NaOH surface treatment or the addition of graphene have also been proposed [36,37,38,39,40]. The incorporation of bioglasses, such Bioglass 45S5 that contains 45 wt% SiO_2_, 24.5 wt% Na_2_O, 24.5 wt% CaO and 6 wt% P_2_O_5_, has attracted significant attention due to its bone-bonding, osteogenic and osteoconductive capabilities [41,42,43]. However, several studies have reported that its incorporation in polymer-based scaffolds decreases the mechanical strength and causes frequent fractures or cracks, which compromises the use of such composite scaffolds for load-bearing applications [44,45,46]. This has been explained by the poor interfacial adhesion, with the bioglass particles acting as defects in the polymer matrix [47]. Moreover, most PCL–bioglass blends are prepared using toxic solvents, such as chloroform and dichloromethane, to dissolve the polymeric material [47,48].

Bone tissue engineering scaffolds have been designed with the aim of achieving optimal biomechanical performance. Usually, scaffolds present pore sizes in the range of 100 to 1000 μm in order to facilitate vascularisation and new tissue formation [49,50,51,52]. However, large pores reduce the overall mechanical properties, making it difficult for cells to bridge. The pore size design that can achieve both optimal mechanical and biological performance is a complex task and has not been fully addressed. Most studies have addressed this issue by assuming that the scaffold’s properties mainly depend on the material composition rather than considering the coupling effect between the composition and the architecture. Previous studies investigating the mechanical performance of PCL scaffolds and PCL scaffolds reinforced with ceramic particles, carbon nanotubes and graphene, presenting uniformly distributed porousness with constant pore size, have found that they do not exhibit the mechanical properties suitable for load-bearing applications [53,54]. This study investigated the behaviour of PCL–Bioglass 45S5 3D printed scaffolds. Blends (10 wt%, 15 wt% and 20 wt% bioglass) were prepared using a simple melt blending approach, which did not require the use of hazardous solvents. Moreover, contrary to previously reported studies that investigated simple rectangular and circular scaffold architectures, this research investigated anatomically designed scaffolds with pore size gradients that mimicked the architecture of bone and thus, aimed to improve the overall mechanical behaviour of the scaffolds and thereby making them suitable for load-bearing applications [55,56,57]. 

## 2. Materials and Methods

### 2.1. Materials

Poly-ε-caprolactone (PCL) (CAPA 6500, Mw = 50,000 Da) was provided by Perstorp Caprolactones (Cheshire, UK) in a pellet shape. Bioglass 45S5 was supplied by CeraDynamics Ltd. James Kent Group (Stoke, UK) in powder form (<10μm particle size). PCL-based composite blends containing different bioglass concentrations (10 wt%, 15 wt%, 20 wt% bioglass) were fabricated using a melt blending procedure. Briefly, PCL pellets were melted at a temperature of 150 °C prior to the addition of bioglass particles at the desired concentrations. The blends were mixed for at least an hour to guarantee a good bioglass dispersion.

### 2.2. Scaffold Fabrication

A screw-assisted extrusion-based additive manufacturing system (3D Discovery system from RegenHU, Switzerland) was used for the fabrication of the PCL–bioglass bone bricks. The 3D Discovery system comprised a material chamber, where the material in a pellet form was melted and pushed into the screw chamber using air pressure (Figure 1). The screw system facilitates the printing process of high viscous materials and guarantees a good mixture in the case of composite materials. Contrary to the commonly used simple 0°/90° or 0°/45° lay-down patterns, a continuous path algorithm was created based on anthropometric measurements using zigzag double filaments (38) and spiral filaments (14), which allowed us to create scaffolds with an overall porosity of 52% (Figure 2 and Appendix A). Scaffolds were fabricated considering the following processing parameters: a melting temperature of 90 °C; the deposition velocity of 20 mm/s; and the screw rotational velocity of 12 rpm. The filaments were extruded using a 0.33 mm diameter needle.

### 2.3. Morphological Characterisation

The scanning electron microscopy (SEM) system FEI ESEM Quanta 200 (FEI Company, Oregon, USA) was used to investigate the morphological characteristics of the 3D printed scaffolds. The samples were sliced with a razor and coated (gold coating) with the use of EMITECH K550X sputter coater (Quorum Technologies, Sussex, UK) before imaging. SEM images were analysed by ImageJ (National Institutes of Health, Bethesda, WA, USA) (10 measurements) to determine the pore size (PS), filament width (FW) and the layers gap (LG).

### 2.4. Water Contact Angle

Surface wettability was determined through water contact angle (WCA) tests, carried out using the OCA 15 system (Data Physics, San Jose, CA, USA). During the tests, deionised water (4 μL of volume drop, 1 μL/s of velocity), in the form of a droplet, was dropped onto the surface of the scaffolds by a fixed pipet, with the scaffolds aligned with the camera, and recorded with a high-speed framing camera. Two time points (0 s and 20 s) were considered and all tests were performed in triplicate, considering three different regions on the scaffolds (Figure 3).

### 2.5. Thermal Gravimetric Analysis

A Netzsch TGA–DTA system (Erich Netzsch GmbH & Co. Holding KG, Selb, Germany) machine was used to determine the material degradation and bioglass concentration in the scaffolds. Tests were conducted in air atmosphere (50 mL/min) with temperatures ranging from 25 °C to 1000 °C at a rate of 10 °C/min. Each test was conducted twice with the use of platinum pans.

### 2.6. X-ray Diffraction

X-ray diffraction (XRD) was used for the crystalline pattern analyses of the PCL and composite bone bricks. Tests were performed using the D2 PHASER Bruker (Bruker Corporation, Billerica, Massachusetts, USA) with a copper anode source. The samples were flattened on a metal holder and data were recorded with a 0.02 2θ° step for a total recording time of 75 min. The detector recording time was 1 s per point. A total of 4500 points were recorded.

### 2.7. Fourier-Transform Infrared Spectroscopy

Fourier-transform infrared spectroscopy (FTIR) tests were conducted using the Nicolet iS 10 system (Thermo Scientific, Altrincham, Manchester, UK) to determine any potential structural changes in both the PCL and composite bone bricks caused by the material preparation and 3D printing process. The transmittance was evaluated considering wave numbers ranging from 3500 to 500 cm^−1^ at room temperature.

### 2.8. Energy Dispersive X-ray Spectroscopy

The chemical composition of the bone bricks, before and after cell seeding, was analysed with the use of Energy dispersive x-ray (EDX) spectroscopy, thereby determining the concentration of calcium (Ca), carbon ©, oxygen (O) and phosphorous (P). The analysis was performed using the SEM FEG FEI Quanta 200 (FEI company, United States). The bone bricks were gold coated prior imaging. The obtained SEM images were analysed using the Oxford AZtec software (Oxford Instruments, Abingdon, Oxford, UK).

### 2.9. Mechanical Characterisation

Compression tests were performed to investigate the influence of the material composition and scaffold architecture. Tests were conducted using the INSTRON 3344 system (Instron, High Wycombe, UK). The scaffolds were compressed with the use of a 2 kN load cell and a 0.5 mm/min displacement rate, according to the ASTM D695-15 standards [58,59]. The dimensions of the bone bricks were 31 mm × 26.7 mm × 10 mm (length × width × height) and tests were repeated four times. Force versus displacement curves that were obtained using the Bluehill Universal Software (Instron, UK) were then converted into stress–strain curves and the compressive modulus was determined using the GraphPad Prism software (GraphPad Software Inc., San Diego, CA, USA) considering the linear portion of the curve.

### 2.10. In Vitro Biological Characterisation

The ability of the printed scaffolds to sustain cell attachment and proliferation was investigated using human adipose derived stem cells (hADSCs) (STEMPRO, Invitrogen, Waltham, Massachusetts, USA) (passage 5). For cell culture, MesenPRO RSTM basal media, 2% (*v/v*) growth supplement, 1% (*v/v*) glutamine and 1% (*v/v*) penicillin/streptomycin (Invitrogen, Waltham, Massachusetts, USA) was used as a nutritional supplement. Cells were harvested at approximately 90% confluency with 0.005% trypsin-ethylenediaminetetraacetic acid (Sigma-Aldrich, Dorset, UK) before cell seeding on the scaffolds.

The scaffolds were sterilised prior to the cell seeding. Briefly, the scaffolds were placed into 50 mL tubes containing 80 wt% of pure ethanol (Thermo Fisher Scientific, Altrincham, Manchester, UK) and 20 wt% of distilled water for 4 h. Then, the liquid was poured out and the scaffolds were washed two times with Dulbecco’s phosphate buffered saline (PBS) (Thermo Fisher Scientific, Altrincham, Manchester, UK) and left to dry for 24 h. Approximately 50,000 cells (counted by Cellometer Auto 100 Bright Field Cell Counter (Nexcelom Bioscience, Lawrence, Massachusetts, USA)) in 89 μL cell suspension were poured onto the top surface of each scaffold with the use of pipettes [60].

Cell viability was investigated at days 1, 7 and 14 after cell seeding using the Alamar Blue assay, which provided a qualitative indication of the metabolic activity of the cells and allowed for indirect information on cell attachment and proliferation on the scaffolds [61,62]. Briefly, at each time point, 90 μL of medium containing 0.001 wt% of Alamar Blue (Sigma-Aldrich, Dorset, UK) was poured in each well and placed in the incubator for 4 h. Then, a certain amount of the sample solution (200 μL) was conveyed into a 96 well plate and the fluorescence intensity was determined using the microplate reader Synergy HTX Multi-Mode Reader (BioTek, Winooski, Vermont, USA) (530 nm excitation wavelength and 590 nm emission wavelength). Then, the scaffolds were washed with PBS and fresh media was added.

On day 14, the morphology of the attached cells and the cell spreading was investigated using SEM imaging. The cells were fixed using 10 wt% neutral buffered formalin (Sigma-Aldrich, Dorset, UK) for 30 min at room temperature and then the scaffolds were rinsed twice with PBS. After the fixation procedure, the scaffolds were dehydrated using graded ethanol mixed with deionised water (50%, 60%, 70%, 80%, 90% and two times with 100%), followed by the addition of 50/50 ethanol/hexamethyldisilazane (HDMS) (Sigma-Aldrich, Dorset, UK) (*v/v*) solution and 100%HDMS, leaving the liquid to evaporate overnight under a hood. Each concentration of ethanol/deionised water and HDMS was used for 15 min.

### 2.11. Data Analysis

Mechanical and biological data are represented as mean ± standard deviation. The statistical analysis was conducted with the use of one-way ANOVA analysis of variance with Tukey’s post hoc test with the use of GraphPad Prism software. Statistically significant differences were considered as * *p* < 0.05, ** *p* < 0.01, *** *p* < 0.001 and **** *p* < 0.0001. TGA, XRD and FTIR data were analysed using Origin 2021 (OriginLab Corporation, Northampton, MA, USA).

## 3. Results and Discussion

### 3.1. Morphological Analysis

3D printing was successfully used to create well-defined scaffolds that presented a gradient of pore sizes, as shown in Figure 4. This figure shows a printed scaffold containing 10 wt% bioglass and the corresponding SEM image of the top, exhibiting a decrease in pore size from the outer region to the internal region of the scaffold. Figure 5 presents the SEM images of the printed scaffolds with different material compositions. The geometrical characteristics are presented in Table 1. As observed, there were some differences between the obtained and designed filament diameter results, which can be attributed to the fact that the same processing conditions were applied to all material compositions. This can be solved by optimising the processing parameters for each material composition. However, such optimisation will result in different melting and cooling processes and, therefore, in different crystallisation conditions with impacts on the biomechanical properties [63]. The results also show that the filament diameter decreased by increasing the ceramic content and, consequently, the pore size increased. This can be explained by the increase in the molten viscosity due to the addition of bioglass. As the processing conditions were constant, the amount of extruded material decreased with the increasing viscosity due to the addition of the reinforcement particles, thus decreasing the filament diameter. Moreover, the SEM images show that the PCL scaffolds presented microporosity at the filament surface (Figure 6A), while the PCL–bioglass scaffolds exhibited a smoother filament surface (Figure 6C,E,G) due to the bioglass concentration, bioglass particle size, recrystallisation process and the crystal size of the bioglass particles [64,65,66,67,68,69,70,71].

### 3.2. Water Contact Angle Analysis

The water contact angle results at two different time points for the different scaffolds that were considered in this study are presented in Figure 6 and Table 2. As observed, the contact angle on a small timescale decreased with time, showing that the scaffolds were absorbing the water or that the water was penetrating through the scaffolds. Moreover, the results at 20 s seem to suggest that the water contact angle slightly decreased from the internal regions of the scaffolds to the external regions. This behaviour, topologically dependent to the scaffold architecture, is due to the gradient increase in the pore size from the internal region to the external region. Furthermore, the addition of bioglass particles into the polymer showed no difference on the overall hydrophilic properties of the scaffolds.

### 3.3. Thermal Gravimetric Analysis

Thermal gravimetric analysis (TGA) showed that the different material compositions exhibited degradation temperatures ranging between 412.67 °C and 437.1 °C (Table 3). After these temperatures, the only remaining content corresponded to bioglass (Figure 7). Moreover, the results show that the addition of bioglass decreased the degradation temperature. This was previously reported by other authors as a consequence of the degradation mechanism between the Si–O^−^ present on the surface of the bioglass particles and the C=O groups on the polymer backbone [72]. The surface modification of the bioglass particles has been proposed to improve the thermal stability of the composite materials [73]. However, considering the processing temperatures used in this study (90 °C), the results suggest that the extrusion process did not induce any material degradation. Moreover, the levels of bioglass present in the scaffolds suggested that the melt blending process was a simple and effective method for the preparation of composite blends without requiring the use of organic solvents to dissolve the polymer.

### 3.4. X-ray Diffraction Analysis

Figure 8 shows the XRD pattern of the PCL and PCL–bioglass 20 wt% scaffolds. In both cases, it is possible to observe strong peaks at 2θ, ranging from 20–24°, which can be associated with the PCL crystal planes (110), (111) and (200). Moreover, the bioglass peaks were similar to the PCL peaks, indicating that the bioglass did not have a crystal structure. According to these results, no chemical transformations were induced during the material preparation and printing process and the bioglass was well-laden in the scaffolds. Moreover, as indicated in Table 4, the full width at half maximum (FWHM) of the PCL–bioglass was lower in comparison to PCL. According to the Scherrer equation [74], the crystallinity and crystallite size decreases with increasing FWHM value, so the addition of bioglass had an impact on the morphological structure of the polymeric matrix and reduced the PCL crystallinity and crystallite size.

### 3.5. Fourier-Transform Infrared Spectroscopy Analysis

FTIR analysis was used to investigate the chemical structure of the filament surface. The results shown in Figure 9 confirm the presence of bioglass particles within the PCL–bioglass scaffolds. For all considered scaffolds, PCL-related stretching modes were observed: asymmetric CH2 stretching peaks at 2949 cm^−1^; symmetric CH_2_ stretching at 2865 cm^−1^; carbonyl stretching at 1727 cm^−1^; C–O and C–C stretching in the crystalline phase 1293 cm^−1^. The asymmetric and symmetric C–O–C stretching can be observed at 1240 cm^−1^ and 1164 cm^−1^, respectively. However, the Si–O–Si bands associated with the bioglass particles cannot be observed due to an overlap with the previously mentioned PCL bands (Figure 9) and the relatively low concentration of bioglass. Nevertheless, the different shape of the PCL peaks at 1164 cm^−1^ in the PCL–bioglass scaffolds can be attributed to the overlap with the Si–O–Si group [75].

### 3.6. Energy Dispersive X-ray Spectroscopy Analysis

Chemical composition analysis was carried out on the PCL and PCL–bioglass scaffolds using EDX spectroscopy to calculate the percentage (wt%) of carbon (C), calcium (Ca), oxygen (O), Silicon (Si) and phosphate (P) (Figure 10). Table 5 shows the element composition on the scaffold’s surface. As observed, the scaffolds containing ceramic bioglass showed less C content, but higher Ca, O, Si and P contents compared to the PCL bone bricks. The results also show that by increasing the bioglass content, the Ca, O, Si and P contents were also increased, confirming the presence of bioglass particles on the filament surface.

### 3.7. Mechanical Characterization Analysis

The design of scaffolds for load-bearing applications is a complex task due to the complex mechanical nature of human bone, which depends on the age, health condition, activity and location (e.g., cortical or compact bone, less porous and presenting high mechanical properties; and trabecular or cancellous bone, presenting high porosity and low mechanical properties) [76,77,78]. Figure 11 shows both the compressive strength and yield strength. The results show that the mechanical properties increased by increasing the bioglass content, highlighting the positive impact of the reinforcement and suggesting a good interface adhesion between the polymeric material and the bioglass particles. This can be attributed to the blend preparation method that, contrary to solvent-based approaches, does not induce any surface changes in the reinforcement particles. Moreover, the pore size gradient allowed us to increase the overall mechanical properties, with all scaffolds (including PCL scaffolds) presenting mechanical properties in the cortical (118–209 MPa compressive strength) bone region [76,77,78].

### 3.8. In Vitro Biological Performance

Figure 12 shows the fluorescence intensity for the different scaffolds at different time points (days 1, 7 and 14) after cell seeding. As high fluorescence intensity values correspond to high cell metabolic activity, the results suggest that the 3D printed scaffolds did not present any cytotoxicity that was able to support cell attachment and proliferation. The results show high cell metabolic activity on the PCL scaffolds at days 1 and 7 in comparison to the PCL–bioglass, which can be attributed to the smooth filament surface of the composite scaffolds that limited cell attachment. This also explains the decrease in metabolic activity when increasing the bioglass content, suggesting low cell attachment and proliferation. However, from day 7 to day 14, a significant increase in cell metabolic activity was observed in the bioglass scaffolds. This can be explained by the improved bioactivity nature of the composite scaffolds, suggesting that bioglass may have a long-term positive biological impact (Figure 13 and Table 6). The results also show that the Ca, O and P contents increased by increasing the bioglass content. This may be attributed to the presence of bioglass acting as a nucleating agent for the precipitation of apatite crystals on the scaffold’s surface [79]. Moreover, Figure 14 shows that the pore size gradient also induced a cell density gradient, with more cells found in the outer regions (large pores, better permeability) and a lower number of cells in the inner regions.

The SEM images of the scaffolds (Figure 15) after 14 days of cell seeding show that the cells were well-spread over the scaffolds (Figure 15A,C,E,G). It is also possible to observe that the number of cells seems to decrease by decreasing the pore size, indicating that the gradient structure could induce cell growth in a gradient manner (cross-section images and clearly indicating the effect of cell density as a function of pore size (Figure 15B,D,F,H). The results also show that cell bridging mainly occurred in the cross-sections of adjacent layers and filaments with higher pore size (Figure 16).

## 4. Conclusions

This paper investigated the coupled effect of scaffold architecture and material composition on the biomechanical performance of PCL–bioglass scaffolds. Scaffolds with pore size gradients ranging from 172 μm to 573 μm were produced using an extrusion-based additive manufacturing system and PCL–bioglass pellets prepared by melt blending. This method avoided the use of organic solvents, which induce chemical changes on the bioglass surface and reduce the adhesion between the polymeric matrix and the reinforcement particles, contributing to a decrease in the mechanical properties. In this study, scaffolds presented mechanical properties in the cortical bone region, with the mechanical performance increasing with increasing the bioglass content. The addition of bioactive glass particles had no significant impact on the hydrophilic characteristics of the bone bricks and, consequently, had no major impact on the initial cell attachment process and the results suggest a positive long-term impact of bioglass. At day 14 after cell seeding, high metabolic cell activity was observed on the PCL–bioglass 10 wt%. This can be explained by the formation of a calcium phosphate layer on the scaffolds that significantly improved bioactivity. Moreover, the results also show that the gradient structure induced cell growth in a gradient manner.

## Figures and Tables

**Figure 1 polymers-14-00445-f001:**
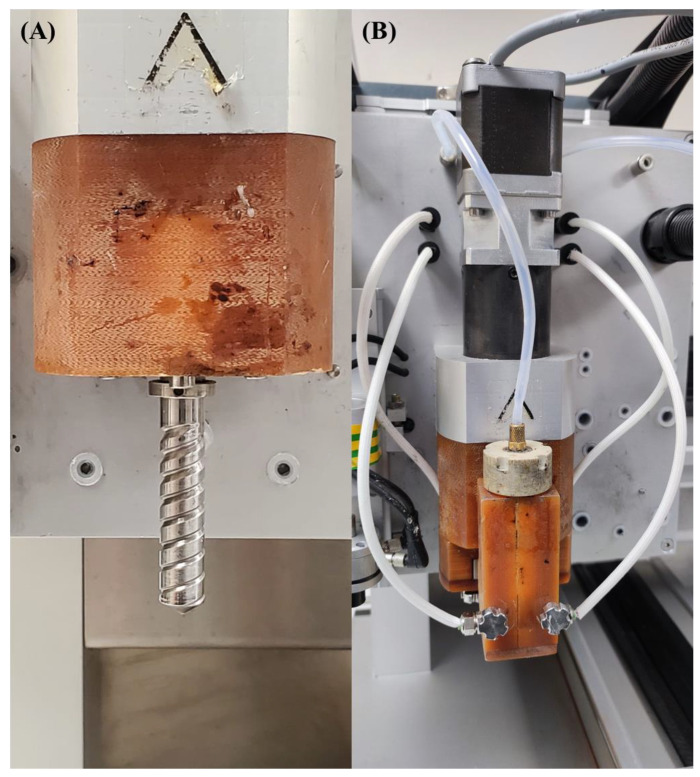
The 3D Discovery system. (**A**) The screw and (**B**) the material chamber and extrusion head.

**Figure 2 polymers-14-00445-f002:**
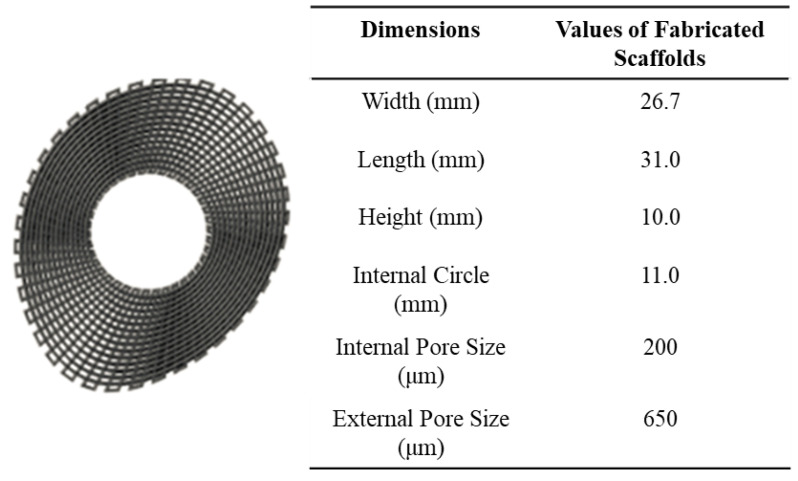
The anthropometric-based geometry and scaffolds dimensions.

**Figure 3 polymers-14-00445-f003:**
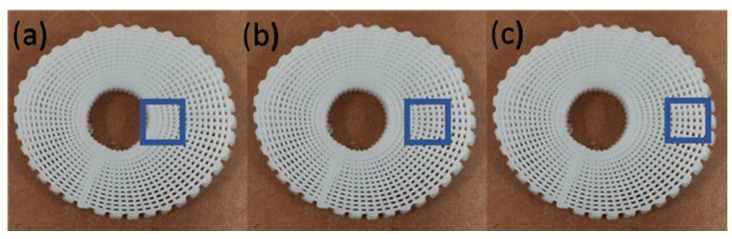
The different regions for the water contact angle tests: (**a**) the internal region; (**b**) the central region; and (**c**) the external region.

**Figure 4 polymers-14-00445-f004:**
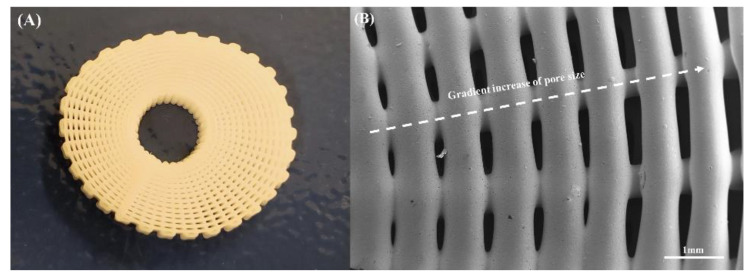
The top view of the 10 wt% bioglass bone bricks: (**A**) The fabricated bone brick and (**B**) the SEM image exhibiting the gradient pore size.

**Figure 5 polymers-14-00445-f005:**
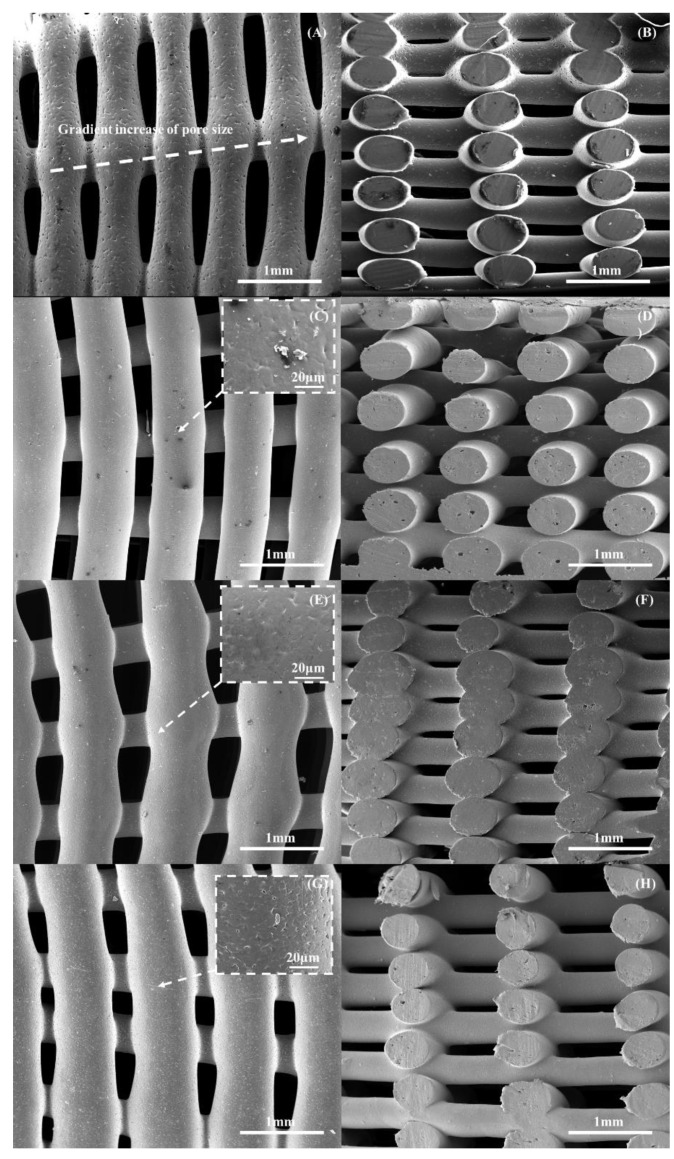
The top and cross-section SEM images of the scaffolds with different material compositions: (**A**,**B**) PCL; (**C**,**D**) bioglass 10 wt%; (**E**,**F**) bioglass 15 wt%; and (**G**,**H**) bioglass 20 wt%.

**Figure 6 polymers-14-00445-f006:**
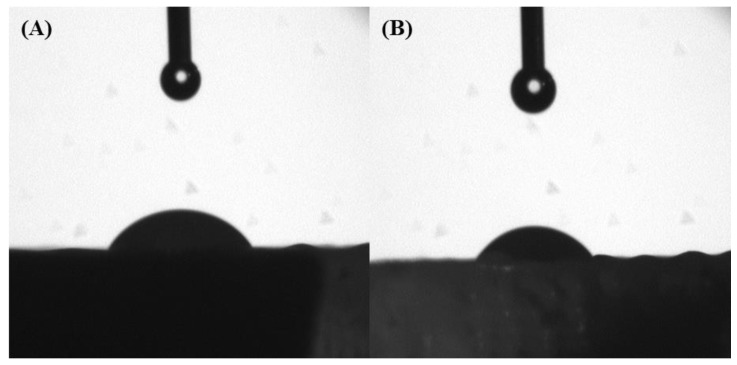
The water drop test on a PCL scaffold filament at 0 s (**A**) and 20 s (**B**).

**Figure 7 polymers-14-00445-f007:**
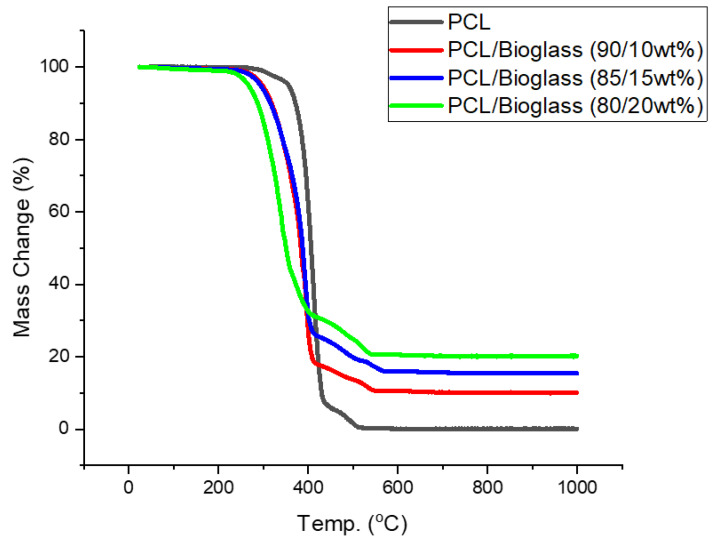
The Thermal Gravimetric Analysis (TGA) curves of the bioglass bone bricks.

**Figure 8 polymers-14-00445-f008:**
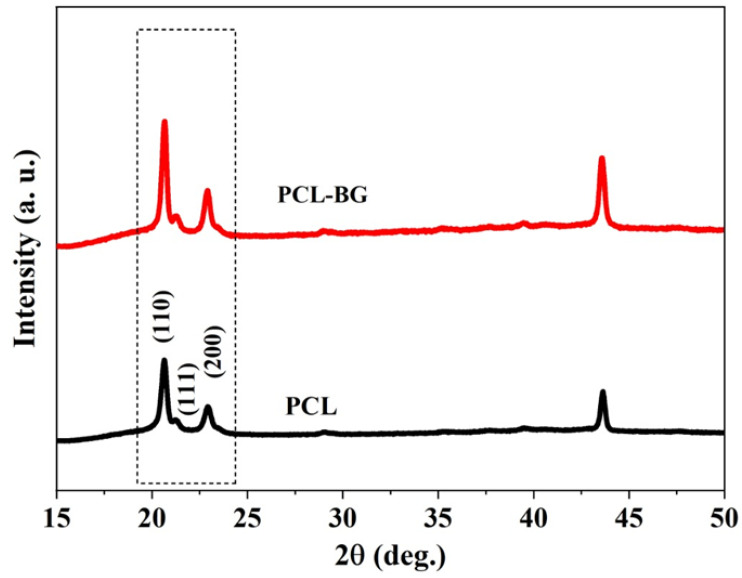
The XRD patterns of the PCL and PCL–bioglass scaffolds (80/20 wt%) in the range of 2θ = 20–24.

**Figure 9 polymers-14-00445-f009:**
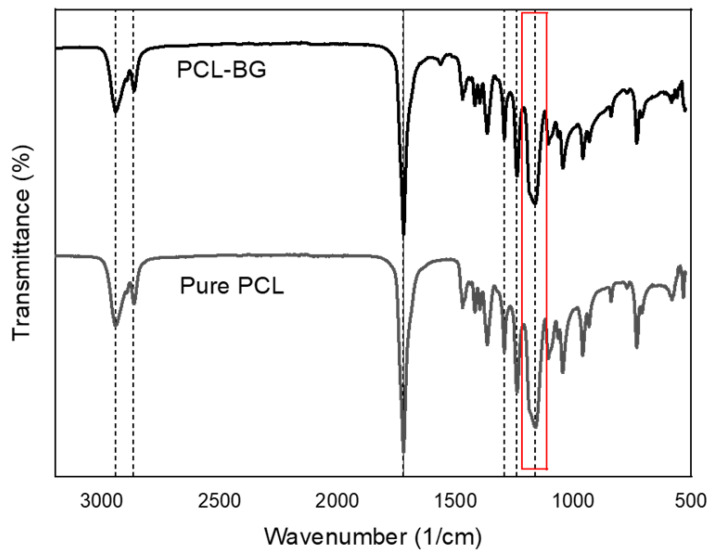
The FTIR spectra of the PCL and PCL–bioglass scaffolds.

**Figure 10 polymers-14-00445-f010:**
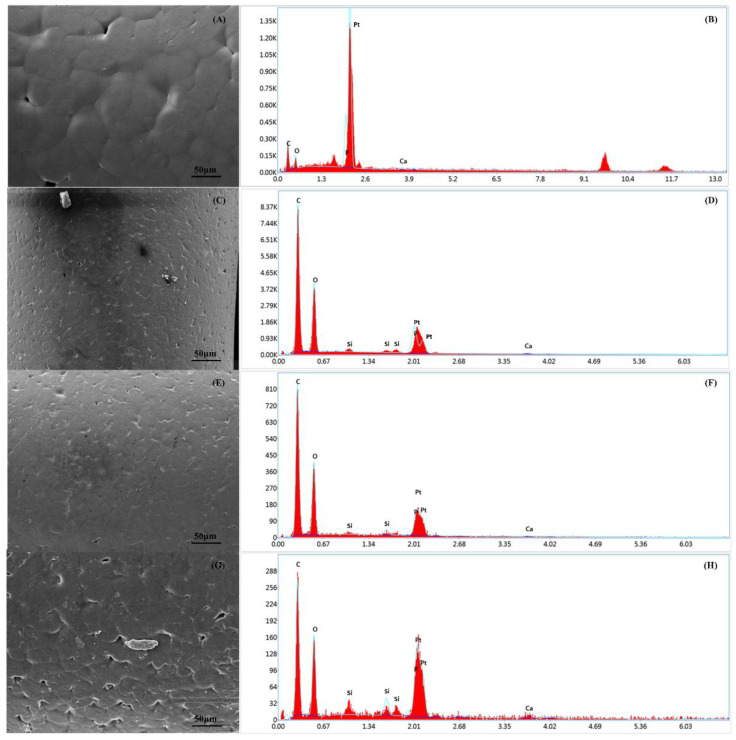
The SEM and EDX spectra of the PCL bone brick (**A**,**B**); the PCL–bioglass 10 wt% scaffolds (**C**,**D**); the PCL/HA 15 wt% scaffolds (**E**,**F**); and the PCL–bioglass 20 wt% scaffolds (**G**,**H**).

**Figure 11 polymers-14-00445-f011:**
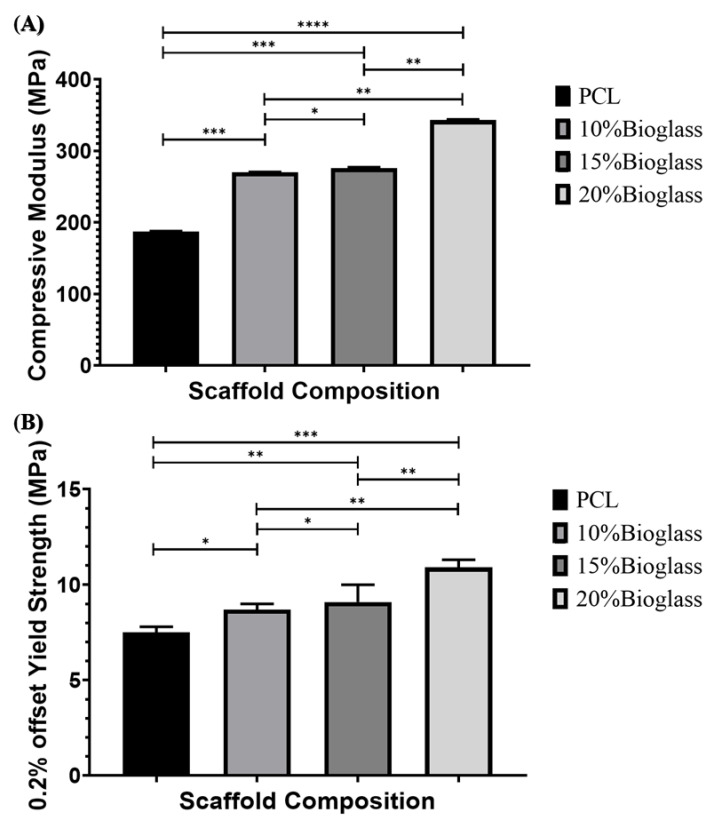
(**A**) The compressive modulus and (**B**) 0.2% offset yield strength results as a function of bioglass content. * Statistical evidence (*p* < 0.05) analysed by one-way ANOVA, and Tukey post hoc test. The * statistical evidence (*p* < 0.05), **, *** and **** is the one-way analysis of variance (one-way ANOVA) and Tukey’s post hoc test with the use of GraphPad Prism software and is used to show the difference between the results. The * is a small difference, while more * are added as the differences between the results increases.

**Figure 12 polymers-14-00445-f012:**
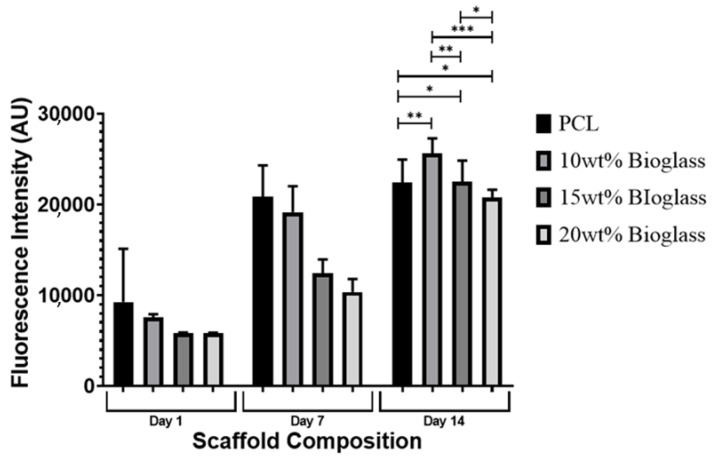
The average fluorescence intensity for the different scaffolds at different days after cell seeding. * Statistical evidence (*p* < 0.05) analysed by one-way ANOVA, and Tukey post hoc test. The * Statistical evidence (*p* < 0.05), ** and *** is the one-way analysis of variance (one-way ANOVA) and Tukey’s post hoc test with the use of GraphPad Prism software and is used to show the difference between the results. The * is a small difference, while more * are added as the differences between the results increase.

**Figure 13 polymers-14-00445-f013:**
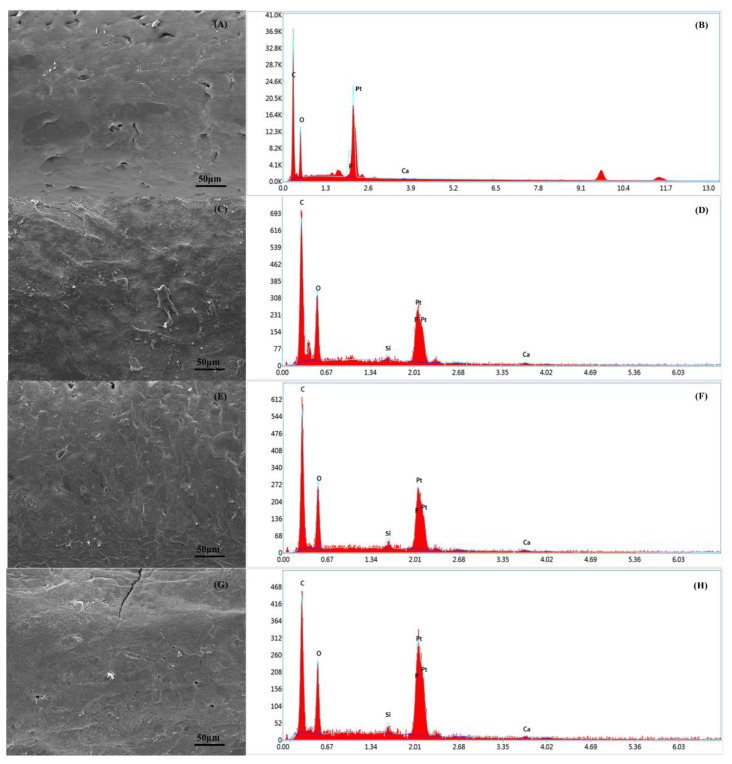
The SEM and EDX spectra of the scaffolds at day 14 after cell seeding: PCL scaffold (**A**,**B**); PCL–bioglass 10 wt% scaffold (**C**,**D**); PCL–bioglass 15 wt% scaffold (**E**,**F**); and PCL–bioglass 20 wt% scaffold (**G**,**H**).

**Figure 14 polymers-14-00445-f014:**
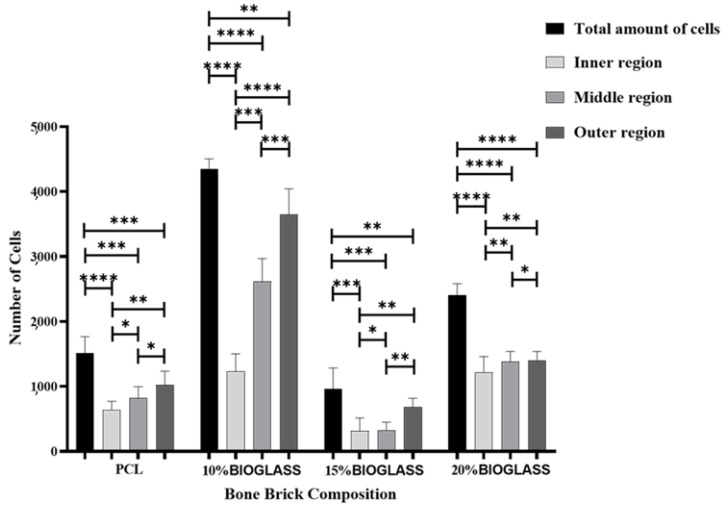
The cell count at day 14 after cell seeding in different regions of the scaffolds. * Statistical evidence (*p* < 0.05) analysed by one-way ANOVA, and Tukey post hoc test. The * statistical evidence (*p* < 0.05), **, *** and **** is the one-way analysis of variance (one-way ANOVA) and Tukey’s post hoc test with the use of GraphPad Prism software and is used to show the difference between the results. The * is a small difference, while more * are added as the differences between the results increases.

**Figure 15 polymers-14-00445-f015:**
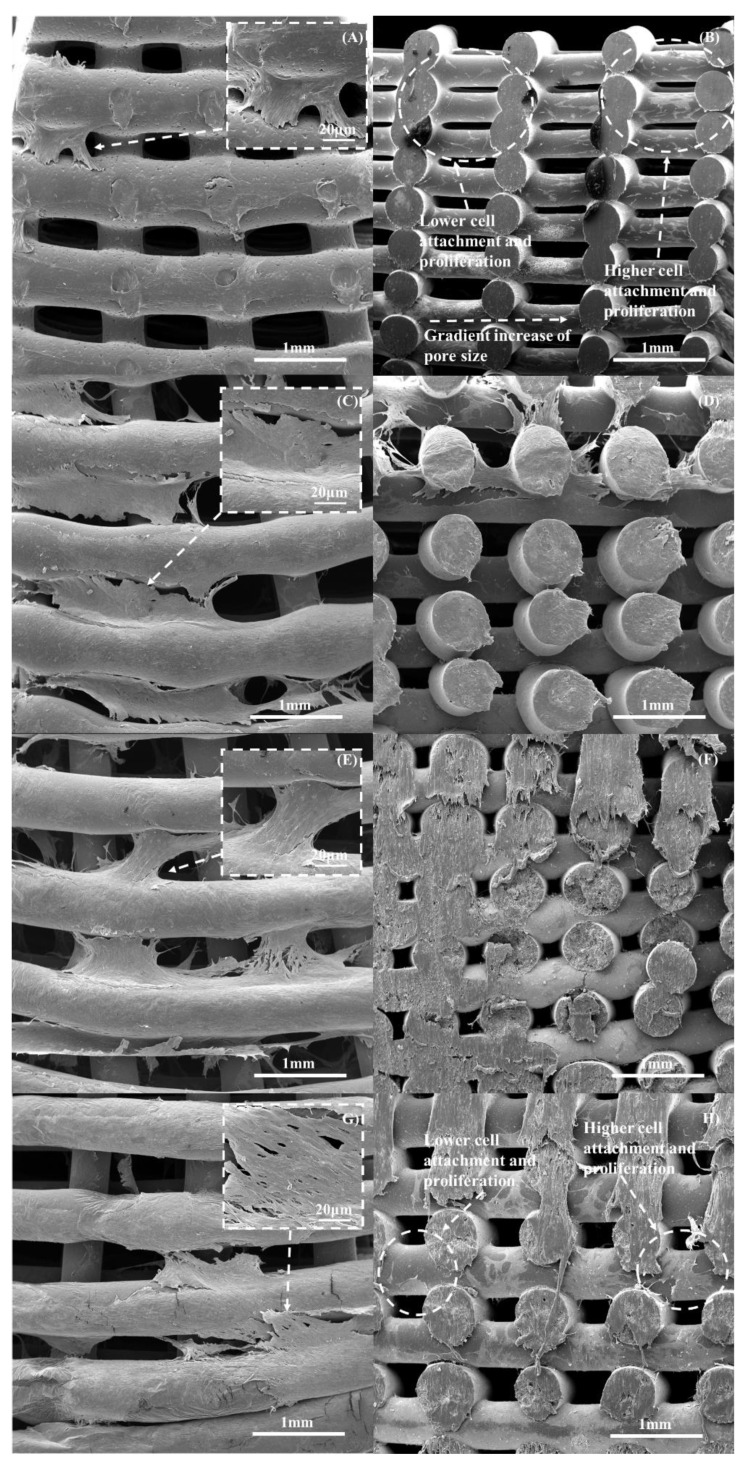
The top and cross-section SEM images showing cell spreading on the scaffolds with different material compositions on day 14: (**A**,**B**) PCL; (**C**,**D**) 10 wt% bioglass; (**E**,**F**) 15 wt% bioglass; and (**G**,**H**) 20 wt% bioglass.

**Figure 16 polymers-14-00445-f016:**
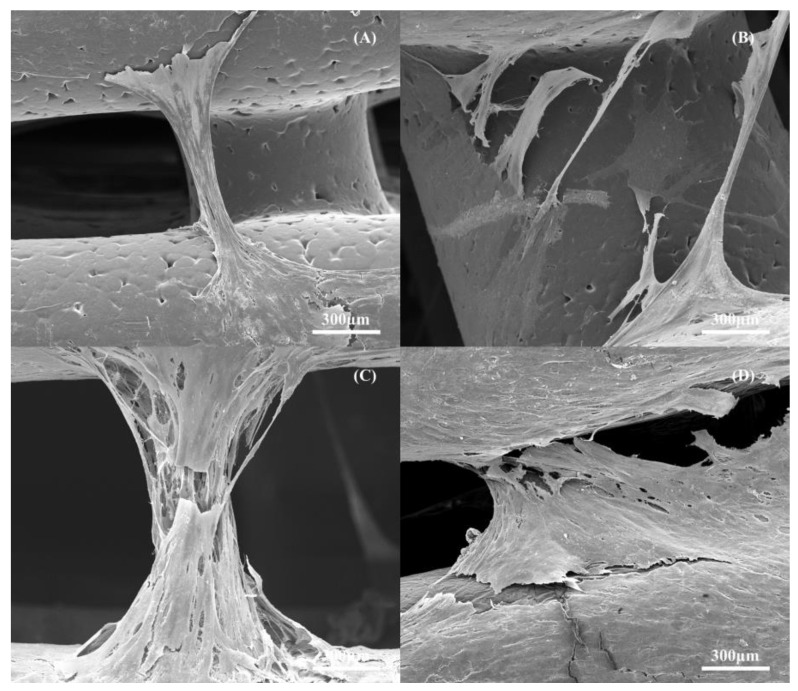
The cell bridging between the 3D printed filaments at day 14 after cell seeding: (**A**) PCL scaffold; (**B**) PCL–bioglass 10 wt%; (**C**) PCL–bioglass 15 wt%; and (**D**) PCL–bioglass 20 wt%.

**Table 1 polymers-14-00445-t001:** The geometric characteristics of the scaffolds.

Material Composition	Pore Size (μm)		Filament Width (μm)
	Internal Region	External Region	
PCL	172 ± 0.01	398 ± 0.08	341 ± 3
PCL/Bioglass (90/10 wt%)	264 ± 0.04	445 ± 0.10	335 ± 11
PCL/Bioglass (85/15 wt%)	289 ± 0.07	541.5 ± 0.09	327 ± 8
PCL/Bioglass (80/20 wt%)	330 ± 0.06	573.4 ± 0.10	325 ± 3

**Table 2 polymers-14-00445-t002:** The WCA results of the different regions of the scaffolds and the different time points.

Material Composition	WCA at 0 s			WCA at 20 s		
	Region a	Region b	Region c	Region a	Region b	Region c
PCL	54° ± 0.5	53° ± 0.7	53° ± 0.9	53° ± 0.3	53° ± 05	53° ± 1.2
PCL/Bioglass (90/10 wt%)	60° ± 0.7	54° ± 0.9	47° ± 1.1	56° ± 0.8	50° ± 0.8	43° ± 1.5
PCL/Bioglass (85/15 wt%)	59° ± 0.3	58° ± 0.6	58° ± 1	56° ± 1.3	56° ± 1.1	55° ± 1.2
PCL/Bioglass (80/20 wt%)	69° ± 0.4	67° ± 0.7	67° ± 0.9	67° ± 0.5	66° ± 0.7	63° ± 1.1

**Table 3 polymers-14-00445-t003:** The designed and manufactured concentrations of the bioglass bone bricks.

Material Concentration	Designed Concentration (wt%)	Measured Concentration (wt%)	Degradation Temperature (°C)
PCL	0	0	437.11
PCL/Bioglass (90/10 wt%)	10	10.61 ± 0.12	415.47 ± 0.26
PCL/Bioglass (85/15 wt%)	15	15.42 ± 0.08	413.93 ± 0.31
PCL/Bioglass (80/20 wt%)	20	20.68 ± 0.05	412.67 ± 0.18

**Table 4 polymers-14-00445-t004:** The calculated PCL crystallinity in the scaffolds and their crystallite size. * The highest peak (crystal plane 110) was used to calculate the crystallinity and crystallite size.

Samples	PCL Crystallinity (%)	FWHM (°) *	Crystallite size (nm) *
PCL	68.9	0.3726	21.7
PCL–Bioglass	59.1	0.2981	20.8

**Table 5 polymers-14-00445-t005:** The element composition of the printed scaffolds.

Material Composition/Element Composition (wt%)	PCL	PCL/Bioglass (90/10 wt%)	PCL/Bioglass (85/15 wt%)	PCL/Bioglass (80/20 wt%)
C (Carbon)	77.65	67.7	66.6	63.6
Ca (Calcium)	0	1.6	1.8	2.1
O (Oxygen)	22.35	19.7	19.8	21
Si (Silicon)	0	10.3	10.8	11.9
P (Phosphate)	0	0.7	1	1.4

**Table 6 polymers-14-00445-t006:** The element composition of the 3D printed scaffolds at day 14 after cell seeding.

Element Composition (wt%)	PCL	PCL/Bioglass (90/10 wt%)	PCL/Bioglass (85/15 wt%)	PCL/Bioglass (80/20 wt%)
C (Carbon)	62.46	49.4	53.3	56.2
Ca (Calcium)	4.2	3.4	2.8	2.5
O (Oxygen)	32.84	31.3	28.5	25.6
Si (Silicon)	0	13.8	14.1	14.8
P (Phosphate)	0.5	2.1	1.3	0.9

## Data Availability

The data presented in this study are available on request from the corresponding author.

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
