# Peer review of "Novel 3D Bioglass Scaffolds for Bone Tissue Regeneration"

_polymers, 2022, doi:10.3390/polym14030445_

Round 1

Reviewer 1 Report

This manuscript entitled "Novel 3D Bioglass scaffolds for bone tissue regeneration" investigates PCL/BG composite scaffolds for bone tissue engineering. After fabrication by melting a mixture of PCL and different BG proportions, the scaffolds are 3D printed with a fine-pored internal structure. 
According to the authors, one innovative feature is the fine-pored internal structure, and the other is the production of the composite by melting.

Bone tissue engineering is a highly researched topic. Various publications have already addressed the topic of PCL/BG scaffolds [1-3]. The materials have been characterized in detail, so few new aspects can be found in the current manuscript.
Thus, I question the publication in such a respected journal as "Polymers". 
Publication is only possible after extensive revision. 

In particular, the innovations should be worked out more clearly. So far, there is not really a unique selling point. Both papers on PCL/BG scaffolds exist see below, as well as publication on innovative internal structures [4]. These should also be discussed. 

Major aspects: 

p. 4, Figure 3
Necessary? I can't really see the point! Should be in supplement data. 

p. 6, l.176 ff
Please insert a reference, or insert the validation of such sterilization!

p.7, l.216
Did not show strand breaks with increasing bioglass percentage? How must the conditions be changed?

p.7, l.228
What explains the smoother surface?

p.8, figure 6 
Is there data on BG distribution in the strand?

p.9 l.239
Data on higher viscosity fluids should be given (blood). The formation of the fracture hematoma is highly relevant for the healing process!

p.14, figure 12
Why was not compared with a normal grid? Should be made up to explain the advantages of the grid?

p. 17 figure 15
The central pore must be included in the discussion. It makes the inner pores more easily accessible. This is no longer to be expected when incorporated into the bone.

  1. Ren, J.; Blackwood, K. a.; Doustgani, A.; Poh, P.P.; Steck, R.; Stevens, M.M.; Woodruff, M. a. Melt-electrospun polycaprolactone strontium-substituted bioactive glass scaffolds for bone regeneration. J. Biomed. Mater. Res. - Part A 2014, 102, 3140–3153.
  2. Poh, P.S.P.; Hutmacher, D.W.; Holzapfel, B.M.; Solanki, A.K.; Stevens, M.M.; Woodruff, M.A. In vitro and in vivo bone formation potential of surface calcium phosphate-coated polycaprolactone and polycaprolactone/bioactive glass composite scaffolds. Acta Biomater. 2016, 30, 319–333.
  3. Dziadek, M.; Pawlik, J.; Menaszek, E.; Stodolak-Zych, E.; Cholewa-Kowalska, K. Effect of the preparation methods on architecture, crystallinity, hydrolytic degradation, bioactivity, and biocompatibility of PCL/bioglass composite scaffolds. J. Biomed. Mater. Res. Part B Appl. Biomater. 2015, 103, 1580–1593.
  4. Söhling, N.; Neijhoft, J.; Nienhaus, V.; Acker, V.; Harbig, J.; Menz, F.; Ochs, J.; Verboket, R.D.; Ritz, U.; Blaeser, A.; et al. 3D-Printing of Hierarchically Designed and Osteoconductive Bone Tissue Engineering Scaffolds. Materials (Basel). 2020, 13, 1836.

Author Response

Dear reviewer,

Kind regards,

Evangelos Daskalakis

Reviewer 2 Report

This is an interesting paper that investigates the effect of pore architecture on PCL/bioglass scaffolds by 3D printing for bone tissue regeneration applications. The work is well guided providing evidence of morphology and properties by experimental techniques such as SEM, water contact angle, TGA, XRD, FTIR and X-ray dispersive energy spectroscopy (for composition). Mechanical characterization is conducted by compression tests and in-vitro biological characterization is carried out providing quantitative evidence of cell viability and proliferation of the composite system. The results are clearly presented and discussed. Conclusions are sound.

Comment for revision:

Page 9-10 TGA

The results provided clearly indicate that the thermal degradation temperature decreases with increasing bioglass content in scaffolds. This is a well- known effect of BG onto polyesters thermal degradation. See for example Larrañaga et al 2014.

The discussion in this point is not sound and should be revised according to the existing literature.

Larrañaga et al Improvement of thermal stability and mechanical properties of medical polyester composites by plasma surface modification of the bioactive glass particles. POLYMER DEGRADATION AND STABILITY 98 (2013) 1717-1723.

Author Response

(The authors gave the same response as above.)

Reviewer 3 Report

Dear Editor and Authors,

This is a paper devoted to studying the effect of scaffold architecture and material composition on the biomechanical performance of PCL/bioglass scaffolds. Several matters should to be considered for publication.

Introduction section

- Line 72-73: "100 to 1000 m” is oddly written. Authors should check again.

Materials and Methods section

- Line 104-105: Authors should check some mistakes such as “Chapter 4”, “filaments (38)” and “spiral filaments (14)”. They were not mentioned in this manuscript.

Results and Discussion section

- Line 251: Table 2: There is a lack of statistical test analysis to assess the significance of the differences.

- Line 243-248: some comments are not consistent with the data (in Table 2). This paragraph should be rewritten.

- Line 287: “1293 cm-1”: typing mistake

Author Response

(The authors gave the same response as above.)

Round 2

Reviewer 1 Report

There is a partial revision of the manuscript. 
Some aspects were not addressed. 
However, since the journal and the manuscript focus on material science characterization and not on biological efficacy or real biological use, the manuscript can be accepted in the current version.